# A novel genetic strategy to interrogate an unknown phenotypic modifier: *Sdhc* KO-Robertsonian mice develop frequent thyroid abnormalities with papillary thyroid carcinoma-like features

**Jean-Pierre Bayley** [1]*, **Heggert G. Rebel**[1], **Peter Devilee**[1,2]

**1** Department of Human Genetics, Leiden University Medical Centre, Leiden, the Netherlands,
**2** Department of Pathology, Leiden University Medical Center, Leiden, the Netherlands

* J.P.L.Bayley@lumc.nl

## Abstract

Genes encoding subunits of the mitochondrial tricarboxylic acid cycle enzyme complex succinate dehydrogenase (SDH) are a leading cause of the neuroendocrine tumour syndrome hereditary paraganglioma-pheochromocytoma. Pathogenic variants of *SDHD* and *SDHAF2* confer a remarkable parent-of-origin tumour risk, in which paternally inherited variants cause tumours but maternally inherited variants do not. Formulated to explain this observation, the Hensen hypothesis proposes that loss of an (unknown) imprinted gene(s), together with the remaining wildtype SDH gene, is a prerequisite for tumour formation; in effect a three-hit hypothesis. This study had three objectives, first, as a test of the Hensen model, second, as a potential model for a disease for which no mouse or cell model currently exists, and finally, as a test of chromosome (Ch.) configuration to interrogate large genomic regions carrying an unknown phenotypic modifier. We crossed a gene knockout line (*Sdhc,* mouse Ch.1) to a Robertsonian chromosome line, Rb(1:7), harbouring the homologous gene imprinting centre (human Ch.11p15, mouse Ch.7) implicated in human tumourigenesis, to create a metacentric chromosome with characteristics of human chromosome 11. We developed 7 cohorts combining *Sdhc* (mouse Ch.1) wildtype or knockout with distinct configurations of Rb(1:7), confirming both paternal and maternal inheritance of *Sdhc*. We noted significant weight gain, and in heterozygote *Sdhc* KO-Rb/wt mice high levels of immune activation. Thyroid abnormalities, including lesions with papillary thyroid carcinoma-like features, were common (30–50%) in *Sdhc* knockout mice with both heterozygous and homozygous Rb chromosomes, regardless of mode of inheritance. We also observed a single case of bilateral pheochromocytoma in which loss of *Sdhc* was not the driver. While our findings did not recapitulate features of the Hensen Model, this study does suggest that chromosomal structure, even in the form of a seemingly innocuous single Robertsonian configuration, can dramatically impact clinical phenotype.

**Data availability statement:** All relevant data are within the manuscript.

**Funding:** Dutch Cancer Foundation (KWF UL 2011-5025) and The Paradifference Foundation.

**Competing interests:** The authors have declared that no competing interests exist.

## Introduction

Paragangliomas and pheochromocytomas/extra-adrenal paragangliomas (PPGL) are closely-related neuroendocrine tumours associated with the parasympathetic and sympathetic nervous systems, respectively. Parasympathetic paragangliomas are most commonly found in the head and neck region, where the most frequent site is the carotid body at the bifurcation of the internal and external carotid arteries. Pheochromocytomas arise in the adrenal medulla and sympathetic paragangliomas in the extra-adrenal sympathetic paraganglia [1]. Most cases of hereditary paraganglioma are attributable to genes encoding subunits of the mitochondrial tricarboxylic acid cycle enzyme succinate dehydrogenase (SDH), including *SDHA, SDHB, SDHC, SDHD*, and the assembly factor gene *SDHAF2*. The SDH protein complex consists of two catalytic subunits, SDHA and SDHB, and two membrane-spanning subunits, SDHC and SDHD, each encoded by a specific gene. *SDHB* and *SDHC* genes are both found, widely separated, on human chromosome 1. *SDHA* is located on human chromosome 5, while *SDHD* is found on human chromosome 11.Unlike pathogenic variants in other SDH subunit genes, variants in *SDHD* and *SDHAF2* (also located on chromosome 11) show a remarkable parent-of-origin dependent tumourigenesis in which tumour formation occurs almost exclusively following paternal transmission of the variant. A large majority of carriers of a paternally inherited variant will develop a tumour at some stage during their lifetime, whereas virtually all maternal carriers will remain tumour-free.

The absence of tumourigenesis in carriers of maternally inherited *SDHD* variants initially suggested a maternally imprinted gene as the underlying cause of the tumour [2–4].However, following identification of *SDHD* in 2000 it was quickly shown that the gene displays biallelic expression in a range of foetal and adult tissues, refuting the idea of direct imprinting of *SDHD*. This and the frequent loss of the entire maternal copy of chromosome 11 in human tumours carrying variants in *SDHD* or *SDHAF2* [5–10] led to the formulation of the 'Hensen model' [11]. This hypothesis proposes that a prerequisite for development of *SDHD*- and *SDHAF2*-related paraglioma, besides a variant in the respective gene, is loss of a paternally-imprinted, maternally-expressed modifier gene likely found in the major gene imprinting centre on human chromosome 11p. Loss of chromosome 11, and especially 11p, is also a characteristic of *VHL*- and some *SDHB*-related tumours [10,12]. It is important to recall that the SDHD and SDHAF2 proteins play disparate roles in SDH complex function. SDHD is a transmembrane subunit, whereas SDHAF2 is an assembly factor that interacts transiently with SDHA during SDH complex assembly. This further supports the idea that gene location rather than protein function is likely the determining factor in the common inheritance pattern. The basis of the Hensen model is that chromosomal location, rather than the specific SDH gene, determines the characteristic pattern of tumour risk.

Extending the Hensen hypothesis to the mouse, we predict that a mouse will develop paraganglioma only when SDH gene function is lost together with the modifier gene located in the equivalent imprinting centre found on mouse chromosome 7. We previously tested one candidate modifier gene (H19) in a cross to an *Sdhd*

KO but that mouse did not develop PPGL [13]. The mouse imprinted region on chromosome 7 contains approximately the same genes as those found in the human imprinting centre. Imprinted genes differ from other genes in that only one parental copy is expressed or predominantly expressed, and most imprinted genes show similar patterns of maternal or paternal imprinting in mice and humans [14]. To develop a Hensen model in the mouse we crossed an *Sdhc* knockout mouse line with a mouse line carrying a Robertsonian Rb(1:7) chromosome.

Rb chromosomes are stable translocations of normally telocentric mouse chromosomes and thus resemble human metacentric chromosomes [15]. Diverse, and often multiple, translocations are found in wild populations of genus *Mus* (incl. the subspecies *Mus musculus domesticus*), some of which have been crossed to laboratory mouse strains and are commercially available. An Rb stain combining *Sdhd* (mouse Ch.9) and the mouse imprinting centre on mouse chromosome 7 was not available, so we instead selected the Rb(1:7) strain which harbours a single Robertsonian chromosome that combines *Sdhc* (Ch.1) with the mouse imprinting centre (Ch.7). As outlined above, we predict that combining functional loss of any SDH gene with loss of the imprinted region may lead to tumour initiation.

The current lack of mouse models for SDH-related paraganglioma is a major obstacle to research. Numerous mouse models have been described over the past two decades [16], but none have developed paraganglioma to date. In view of the Hensen model, an unknown gene (or genes) playing an essential permissive role in human PPGL may be the missing piece in the puzzle of why mice fail to develop SDH-related paraganglioma. This approach serves a triple function as evidence supporting or refuting the Hensen model, as a potential paraganglioma model, and as a test of the use of chromosomal configuration to interrogate large genomic regions carrying an unknown phenotypic modifier.

## Methods

### Ethics statement

This study was approved by the Animal Experimentation Committee of Leiden University Medical Center (DEC nr12069).

### Cohort generation

The C57BL/6N-*Sdhc*Tm1a(EUCOMM) Wtsi mouse, obtained from the Sanger Center, UK, on behalf of the European Mutant Mouse Archive, was developed by the European Conditional Mouse Mutagenesis (EUCOMM) Program and has been extensively phenotyped by the International Mouse Phenotyping Consortium (IMPC), which reported homozygous embryonic lethality, as well as increased body weight and thrombocytosis in heterozygotes. These animals carry a LacZ cassette inserted into intron 3 of *Sdhc* that causes skipping of exon 4. Skipping in the provided animals was confirmed by RT-PCR, and cross breeding ratios of heterozygote mice indicated homozygote lethality. The Rb mouse line, B6Ei.Cg-Rb(1.7)1Rma/J [Universita di Roma Rb(1:7)] was obtained from the Jackson Laboratory. This Rb strain carries a single translocation of chromosome 1 and chromosome 7.

Male mice carrying a heterozygous Rb chromosome [Rb/wt] were initially bred to wild type C57BL/6N females, and mice heterozygote for the Rb chromosome were then interbred to obtain heterozygote [wt/Rb] and homozygote mice [Rb/Rb]. Mouse cohorts were generated by initial crossing to ingress the *Sdhc* KO allele to the Rb background: *Sdhc+/-* x Rb1:7(*Sdhc+/+*)/Rb Rb1:7(*Sdhc+/+*), resulting in *Sdhc-*/Rb*Sdhc+* mice. *Sdhc-*/Rb1:7(*Sdhc+*) mice were then crossed with Rb1:7(*Sdhc+*)/Rb Rb1:7(*Sdhc+*) mice and bred as available. Taking into account known transmission distortion in Rb mice (around 10%) and high recombination rates due to chromosomal distance, we expected an efficiency of ~10% to generate Rb1:7(*Sdhc-*)/Rb1:7(*Sdhc+*) mice. Subsequent crosses of Rb1:7(*Sdhc-*)/Rb1:7(*Sdhc+*) x Rb1:7(*Sdhc-*)/Rb1:7(*Sdhc+*) generated further experimental and control cohorts. Mice were genotyped by PCR for the presence of the mutant *Sdhc* allele and results were compared to metaphase spreads (using standard protocols) from the same animals to determine combined genetic and chromosomal status.

## Animals and pathology

Mice were managed according to guidelines of the the Animal Experiment Committee of Leiden University Medical Center (DEC nr.12069), and housed under standard conditions (temperature of 22±1.5 °C, 12-h light/12-h dark cycle) with constant access to food and water. (1) Methods of sacrifice: Mice showing signs of distress were sacrificed humanely in a $CO_2$ chamber. (2) Efforts to alleviate suffering: Mice were monitored twice weekly by investigators, and twice weekly by animal facility staff. Mice showing signs of distress were sacrificed and investigated for signs of general disease in all organs, with a special focus on paraganglioma-related areas including the carotid bifurcation and the adrenal gland. The experiment was terminated at approximately two years. Mouse pathology was assessed by the investigators, by an experienced human pathologist and by an experienced animal pathologist. Tumours and selected pathological tissues were snap frozen and preserved as formalin-fixed paraffin-embedded (FFPE) tissue.

## (Immuno)histochemistry

Haematoxylin/eosin staining of FFPE sections was carried out using standard methods. Paraffin sections were prepared for immunohistochemistry as follows: antigen retrieval in 10 mM TRIS/1 mM EDTA buffer, pH 9.0, for 10 minutes in a microwave, blocking of endogenous peroxidase with methanol/hydrogen peroxide (95%/5%, v/v), then PBS with 5% bovine serum albumin to block aspecific antibody binding, triple washes in PBS/Tween (0.05%), then incubated overnight with the primary antibody (see below) diluted in PBS/1% skimmed milk. The next day, slides were washed in PBS/Tween 0.05% and incubated with Envision+ (DAKO Agilent, Middelburg, The Netherlands) and the chromogen 3,3′-diaminobenzidine (DAB) according to manufacturer's instructions. After washing in deionized water, slides were briefly counterstained with haematoxylin (Klinipath, Duiven, Netherlands) and directly rinsed in running tap water. Primary antibodies for IHC analysis were as follows: anti-thyroglobulin (GA509, DAKO Agilent, Middelburg, The Netherlands; 1:1000), anti-tyrosine hydroxylase (ab112, Abcam, Amsterdam, The Netherlands; 1:2000), anti-SDHB (HPA002868, Sigma Atlas, BioConnect, Huissen, The Netherlands; 1:4000) and anti-5-hydroxymethylcytosine (5hmC) (39770, Active Motif Europe, Waterloo, Belgium,; 1:7000).

## Statistics

Standard descriptive statistics were used to analyse data, including Student's T test and 2x2 contingency tables where appropriate (Graphpad, USA). All outcomes were corrected for multiple testing.

## Results

### Experimental cohorts

To approximate the Hensen model in the mouse we crossed an *Sdhc* knockout mouse line with a mouse line carrying a Robertsonian (Rb) chromosome [Rb(1:7)] (Fig 1A and 1B). Human and mouse chromosomal configurations differ but certain large regions show high homology, including the imprinting centre on human chromosome 11p and mouse chromosome 7 (Fig 1C and 1D). The Rb(1:7) line carries a single Robertsonian chromosome that unites mouse *Sdhc* and the imprinting centre on mouse Ch.7. This murine 'Hensen model' depends on spontaneous loss of the entire maternal copy of a Robertsonian chromosome carrying the remaining wildtype copy of Sdhc and the mouse imprinting centre, a signature event in human *SDHD* and *SDHAF2*-linked paragangliomas [10].

We developed a total of 7 cohorts involving paternal and maternal transmission of *Sdhc*, and heterozygous and homozygous Rb(1:7) configurations (Fig 2). These included three control cohorts: (1) *Sdhc+/-* on the BL6 background (no Rb), (2) an *Sdhc+/+* line with a single Rb chromosome, and (3) an *Sdhc+/+* line with two Rb1:7 chromosomes. The experimental cohorts (4–7) all included *Sdhc+/-* with either one or two Rb chromosomes (to address possible differences in chromosomal stability). These crosses ensured inheritance of *Sdhc* via either the female (4&6) or male (5&7) line, and

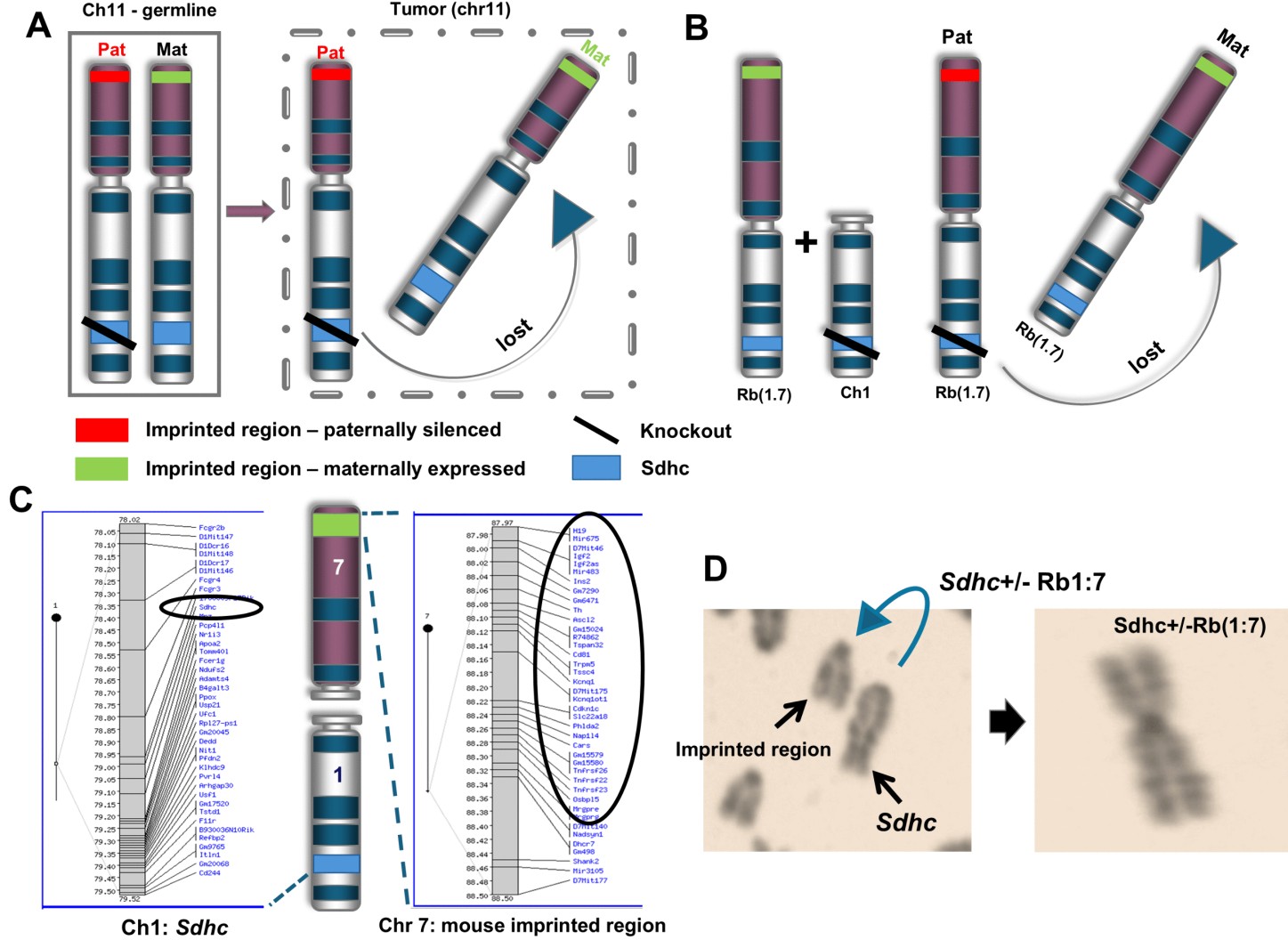

**Fig 1. The 'Hensen model' in outline.** (A) Loss of an imprinted gene in a human paraganglion cell, together with the normal gene copy of SDHD, leads to tumour formation (Pat – paternal, Mat – maternal) (B). The 'Hensen model' in the mouse. After recombination with a telocentric mouse chromosome 1 carrying the Sdhc knockout, the Rb1:7 Robertsonian chromosome carries a heterozygous mutated copy of the PPGL tumour suppressor gene Sdhc on chromosome 1 (blue) (slash indicates knockout) and the mouse equivalent of the human imprinting centre on human chromosome 11 (green, mouse chromosome 7). The scenario (right) depicts Sdhc+/-, Rb(1:7)/Rb(1:7) [Cohorts 6 & 7] but we also developed Sdhc+/-, Rb(1:7)/wt [Cohorts 4 & 5] (see Fig 2) (C) Human chromosome 11 and mouse chromosome 7 share twenty genes that are (https://www.geneimprint.com/site/genes-by-species) reportedly imprinted in one or both species (D). An Rb(1:7) mouse metaphase spread with normally telocentric chromosomes (left) includes a Robertsonian chromosome (right).

the latter cohorts carrying a paternally inherited *Sdhc* KO (5&7) simulate patients at risk of tumour development due to a paternally inherited *SDHD* or *SDHAF2* variant (Fig 1A). This genetic configuration uniting *Sdhc* (Ch.1) and the imprinting centre (Ch.7) loosely models human chromosome 11, with the caveat that other intervening chromosomal regions differ from human chromosome 11. As tumour development in this *Sdhc*-Rb model is likely dependent on the spontaneous loss of the normal copy of the chromosome, we generated and followed a large experimental group of 126 mice, including 80 in the paternally inherited cohorts. As clinical detection of PPGL in humans typically occurs around the ages of 35–40 years, we followed these mice until aged (mean 70 weeks, range 17–152 weeks).

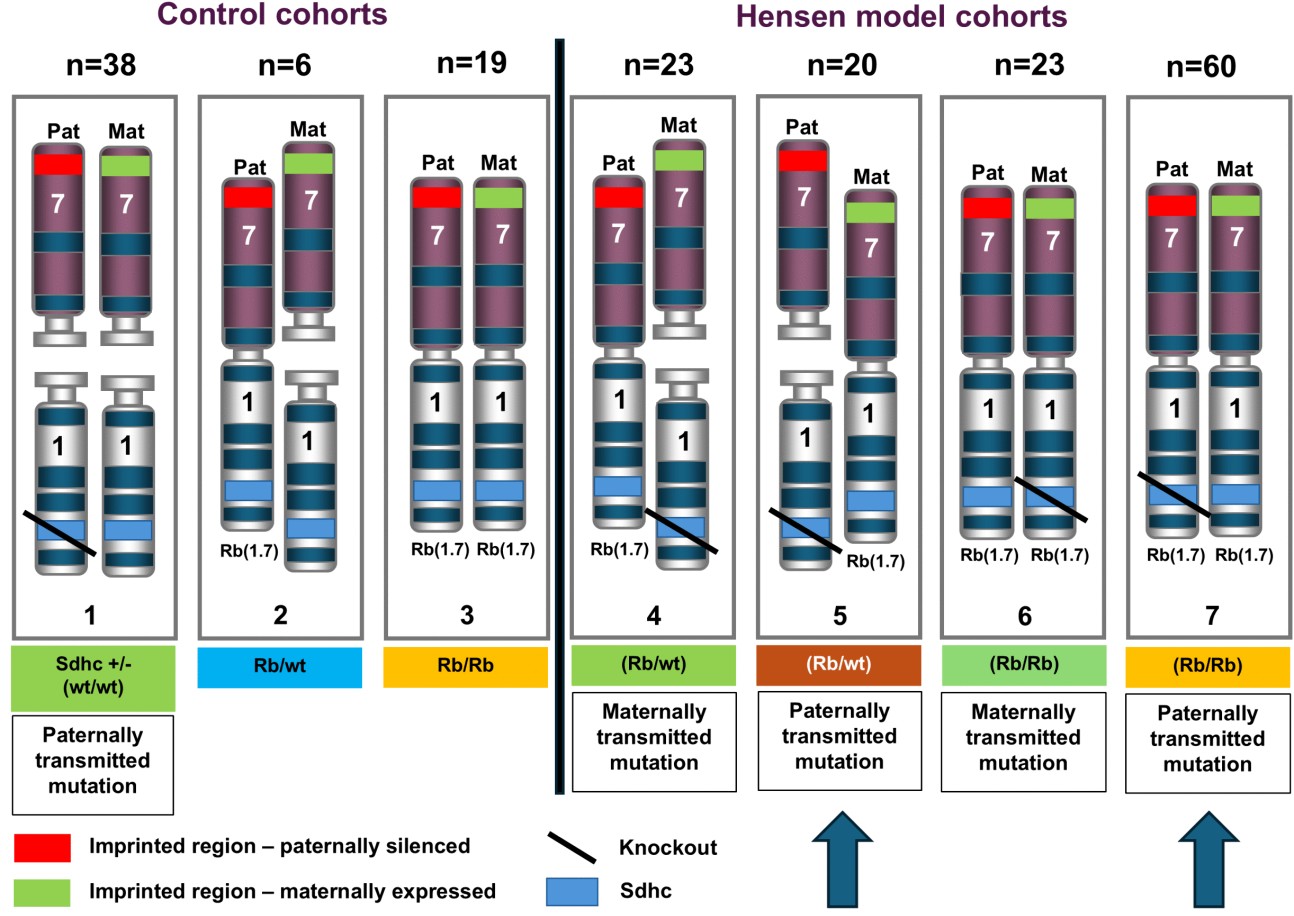

**Fig 2. Testing the Hensen model in the mouse.** Sdhc+/- Rb1:7 experimental cohorts in outline (numbered 1 to 7). These cohorts were designed to model the parent-of-origin inheritance effect in the mouse. Cohorts 1, and 4 to 7 all carry a heterozygous knockout (null allele) of Sdhc. According to this model, both functional gene copies (Sdhc+unknown imprinted modifier) should be lost following an LOH event, leading to tumorigenesis. Cohorts 5 and 7 had a paternally inherited Sdhc KO (arrows), and thus most closely resembled the Hensen model (in humans). Cohorts 4 and 5 were created to assess possible differences in chromosomal stability due to the configuration of Robertsonian chromosomes. The cohorts with maternally-inherited mutations acted as controls for the Hensen model (Mat = maternal, Pat = paternal).

## Mouse body weight

The loss of *Sdhc* is associated with weight gain in mice, as previously reported by the IMPC and a study using a conditional Sdhc KO [17]. The IPMC reported weights of 43g and 33g for males and females, respectively, (n = 15) in heterozygous *Sdhc+/-* mice. In our cohort *Sdhc+/-* mice showed similar weights of 44g and 41g for males and females, respectively (n = 17) (Fig 3). Interestingly, mice in the experimental cohorts 4 and 5, carrying a heterozygous *Sdhc* KO and a single Rb chromosome, showed the highest weights compared with both the control cohorts (All Coh 1–3 versus All Coh 4&5, **p > 0.0001) and the Rb/Rb experimental cohorts (All Coh 4&5 versus Coh 6&7, p = 0.005). Nevertheless, cohorts 6&7 were also significantly heavier compared with cohorts 1–3 (p = 0.03). In cohorts 4&5 versus 6&7, males accounted for most of the excess weight (males p = 0.01; females p = NS). In the experimental cohorts, carrying a single Rb chromosome together with heterozygosity for *Sdhc* appeared to confer excess body weight in addition to the contribution of *Sdhc* alone (Fig 3).

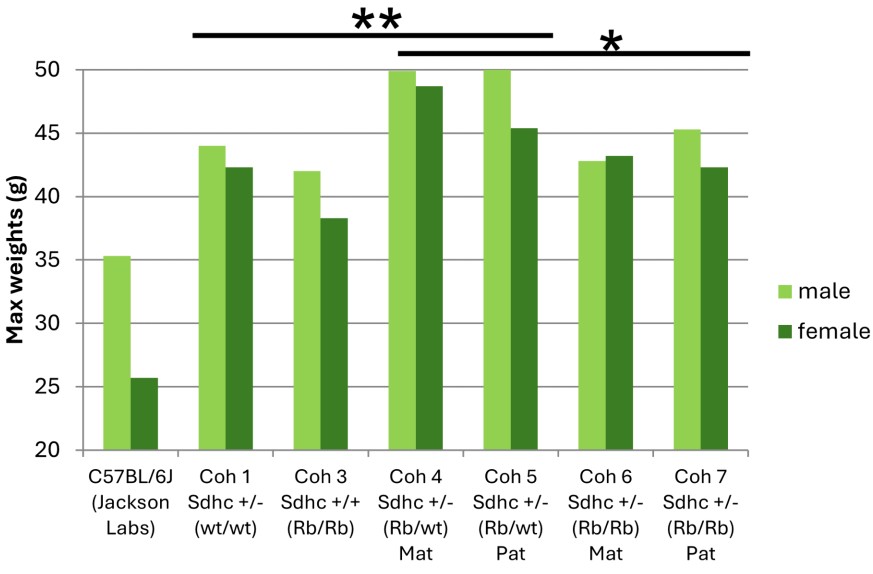

| Number and max weight (gr) | | | | |
|---|---|---|---|---|
| | male | | female | |
| C57BL/6J (Jackson Labs) | NA | 35.3g | NA | 25.7g |
| Coh 1 *Sdhc* +/- (wt/wt) | 9 | 44g | 7 | 42.3g |
| Coh 3 *Sdhc* +/+ (Rb/Rb) | 17 | 42g | 12 | 38.3g |
| Coh 4 *Sdhc* +/- (Rb/wt) Mat | 20 | 49.9g | 13 | 48.7g |
| Coh 5 *Sdhc* +/- (Rb/wt) Pat | 7 | 50g | 8 | 45.4g |
| Coh 6 *Sdhc* +/- (Rb/Rb) Mat | 20 | 42.8g | 13 | 43.2g |
| Coh 7 *Sdhc* +/- (Rb/Rb) Pat | 31 | 45.3g | 35 | 42.3g |

**Fig 3. Maximum recorded weight of the various cohorts.** Weight of wildtype C57BL/6J (data Jackson Laboratory – left) at 25 weeks included for reference. Mean cohort age was 85 weeks. Cohorts 4 and 5, carrying a heterozygous Sdhc KO and a single Robertsonian chromosome, showed the highest weights (All Coh 4-5 versus Coh 6-7, two-tailed unpaired T test *p = 0.005, corrected; All Coh 1-3 versus All Coh 4-5, **p > 0.0001). In cohorts 4&5 versus 6&7, males accounted for most of the excess weight (males p = 0.01, corrected; females p = NS). Although lower than cohorts 4&5, cohorts 6&7 carrying two Robertsonian chromosomes were significantly heavier compared with cohorts 1-3 (p = 0.03, corrected). To avoid confounding effects of pathology on body weight, weights were analyzed as maximum lifetime weight achieved (mean number of times weighted 11.4, range 1-14). P values corrected for multiple testing.

## Pathology

In addition to the adrenal gland, the abdominal aortal area and carotid body, the primary sites of paraganglioma-pheochromocytoma in humans, we also assessed general pathology in each cohort (Fig 4). A low level of pathology in various organs is common in aged mice. Across all cohorts pathology was broadly similar, with the exception of liver, immune and thyroid abnormalities. Liver pathologies appeared least often in cohorts carrying two copies of the Rb chromosome (cohorts 3, 6 & 7).

## Inflammation

A second difference between cohorts was the regular occurrence of gross, systemic inflammation in mice carrying a single copy of a Rb chromosome (Fig 4). This was apparent due to grossly enlarged lymph nodes and/or spleen (Fig 5). While this might seem related to skin injuries, self-inflicted or due to fighting, which were broadly more common in experimental versus control cohorts, inflammation did not correlate with either skin injury or with barbering (without skin injury).

| Mouse cohorts | Number of mice | Liver (N, %) | Lung (N, %) | Ovary-Uterus (N, %) | Prostate (N, %) | Eye (N, %) | Kidney (N, %) | Intestine-mesent (N, %) | Inflam (N, %) | Thyroid (N, %) | Adrenal-Carotid body (N, %) | Salivary (N, %) | Skin-bitten (N, %) | Barbering (N, %) |
|---|---|---|---|---|---|---|---|---|---|---|---|---|---|---|
| Cohort 1 Sdhc +/- (wt/wt) | 38 | 8 (21.1) | 4 (10.5) | 3 (7.9) | 0 (0.0) | 0 (0.0) | 3 (7.9) | 1 (2.6) | 2 (5.3) | 2 (5.3) | 6 (15.8) | 0 (0.0) | 1 (2.6) | 4 (10.5) |
| Cohort 2 Sdhc +/+ Rb/wt | 6 | 2 (33.3) | 1 (16.7) | 0 (0.0) | 0 (0.0) | 0 (0.0) | 1 (16.7) | 2 (33.3) | 3 (50.0) | 1 (16.7) | 1 (16.7) | 1 (16.7) | 1 (16.7) | 0 (0.0) |
| Cohort 3 Sdhc +/+ Rb/Rb | 19 | 0 (0.0) | 1 (5.3) | 3 (15.8) | 0 (0.0) | 2 (10.5) | 1 (5.3) | 0 (0.0) | 2 (10.5) | 3 (15.8) | 3 (15.8) | 0 (0.0) | 2 (10.5) | 3 (15.8) |
| Cohort 4 Sdhc +/- (Rb/wt) | 23 | 7 (30.4) | 1 (4.3) | 1 (4.3) | 0 (0.0) | 3 (13.0) | 1 (4.3) | 6 (26.1) | 12 (52.2) | 9 (39.1) | 2 (8.7) | 0 (0.0) | 4 (17.4) | 0 (0.0) |
| Cohort 5 Sdhc +/- (Rb/wt) | 20 | 2 (10.0) | 2 (10.0) | 2 (10.0) | 0 (0.0) | 3 (15.0) | 0 (0.0) | 3 (15.0) | 6 (30.0) | 10 (50.0) | 5 (25.0) | 0 (0.0) | 4 (20.0) | 6 (30.0) |
| Cohort 6 Sdhc +/- (Rb/Rb) | 23 | 2 (8.7) | 1 (4.3) | 1 (4.3) | 0 (0.0) | 1 (4.3) | 2 (8.7) | 4 (17.4) | 2 (8.7) | 7 (30.4) | 3 (13.0) | 1 (4.3) | 8 (43,0) | 6 (26.1) |
| Cohort 7 Sdhc +/- (Rb/Rb) | 60 | 2 (3.3) | 9 (15.0) | 4 (6.7) | 1 (1.7) | 4 (6.7) | 0 (0.0) | 5 (8.3) | 9 (15.0) | 26 (43.3) | 6 (10.0) | 5 (8.3) | 13 (21.7) | 7 (11.7) |

| Key | |
|---|---|
| Liver | Liver tumors/cysts |
| Lung | Lung disease including tumors |
| Ovary-Uterus | Ovarian and uterine dsease, inlcuding ovarian haemangioblastomas |
| Prostate/Sem Ves | Enlarged prostate / Seminal vesicles |
| Eye | Eye disease/blindness |
| Kidney | Kidney disease |
| Intestine-mesenteric | Intestinal-mesenteric pathology incl. lymphomas |
| Inflammation | Enlarged lymph nodes and spleen |
| Thyroid | Enlarged thyroid |
| Adrenal-Carotid body | Enlarged adrenal glands and/or carotid body |
| Salivary | Enlarged salivary glands |
| Skin-bitten | Skin problems, injury due to fighting |
| Barbering | Barbering (cropping of hair) |

**Fig 4. Overview of gross phenotypes and frequencies found in Sdhc +/- Rb1:7 cohorts.** Columns indicate organs visually assessed by investigators for signs of pathology. Key provides more detail on pathology observed. Liver tumours/cysts (not further investigated) were marginally more common in cohorts 4 & 5. Inflammation refers to noticeable enlargement of any lymph node and/or spleen, which was significantly more frequent in cohorts 4 & 5 harbouring one copy of the Rb1:7 Robertsonian chromosome (p = 0.007). The main finding, beside an incidental bilateral pheochromocytoma, was conspicuous pathology of the thyroid, which was highly significant in the experimental versus control cohorts (Coh 1-3 vs. Coh 4-7, p = 0.0001). P values corrected for multiple testing (13 tests).

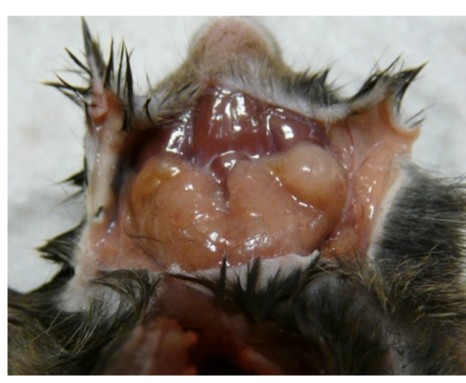

Enlarged mandibular or parotid lymph node

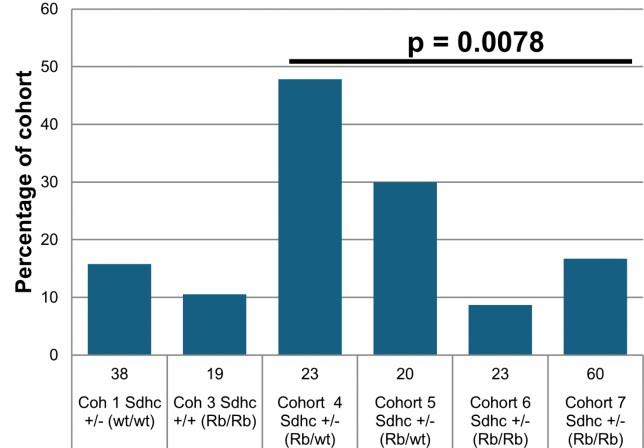

**Fig 5. Inflammation in heterozygous Rb mice.** Signs of inflammation were significantly more common in Sdhc +/- Rb1:7 cohorts carrying only one Robertsonian chromosome compared to cohorts with two or no Robertsonian chromosomes. Inflammation in this cohort did not correlate with fighting or other external injury or with any other pathology. Bars show percentage of each cohort. Numbers of mice shown below bars. Cross-cohort Chi Sq (corrected) significance p = 0.02. Cohorts 4&5 versus cohorts 6&7, 2x2 contingency p = 0.007 (corrected). Corrected = correction for multiple testing (13 tests).

## Thyroid abnormalities

Gross pathology of the thyroid is extremely rare in the C57BL/6N mouse line and was uncommon in *Sdhc+/+* line (Cohort 1; Figs 4 and 6). However, the experimental *Sdhc+/-* Rb1:7 cohorts developed high rates of thyroid disease (Figs 4 and 6), with mice exhibiting hyperplasia and lesions with papillary thyroid carcinoma (PTC)-like features. Up to 50% of mice in the Rb experimental cohorts exhibited enlarged thyroid glands, which emerged from under the sternothyroid muscle upon necropsy. The low level of abnormalities in the *Sdhc+/-* cohort and the higher rate in the Rb/Rb (*Sdhc+/+*) cohort suggest a possible synergistic effect of one or more Rb chromosomes with the *Sdhc* KO. Histochemical analysis of the thyroid of affected mice showed extensive hyperplasia, cystic areas and a predominantly papillary morphology in many cases, reminiscent of PTC (Figs 7–8).

## Paraganglioma and pheochromocytoma

None of the cohorts, including those that inherited *Sdhc* via paternal transmission (cohorts 5&7), developed paraganglioma or frequent pheochromocytoma. Nevertheless, an interesting incidental finding was bilateral pheochromocytoma in a single mouse (9100515 *Sdhc+/-* female, cohort 5 or 7 – metaphase unsuccessful). The left-sided tumour showed dramatic enlargement of the adrenal medulla (Fig 9A and B), consistent with a diagnosis of pheochromocytoma (643 mg). The right-sided adrenal was also grossly enlarged (Fig 9B) at 18 mg, which is 5x normal mean weight. The mean size of the adrenal gland in wildtype C57BL6 is 3.6 mg (range 1 mg to 5.5 mg) (Fig 9C). Immunohistochemical (IHC) staining with anti-tyrosine hydroxylase, a marker protein found in the adrenal medulla and paraganglioma/pheochromocytoma, showed histology typical of pheochromocytoma (Fig 9D). To assess a possible role for loss of *Sdhc* in the genesis of these tumours, we carried out anti-SDHB IHC, which is usually negative in SDH-associated tumours due to loss of the SDH protein complex. Here, however, staining was positive, suggesting no role for *Sdhc* loss (Fig 9E). Another characteristic of SDH-deficient tumours is loss of 5-hydroxymethylcytosine (5hmC), as illustrated by a human SDHB-deficient tumour (Fig 9F). Pheochromocytoma cells from mouse 9100515 showed anti-5hmC staining (Fig 9G) comparable to a wildtype mouse (Fig 9H), suggesting retention of 5hmC and again no causative link to loss of *Sdhc*. Cells from the mouse pheochromocytoma were cultured but failed to proliferate (Fig 9I).

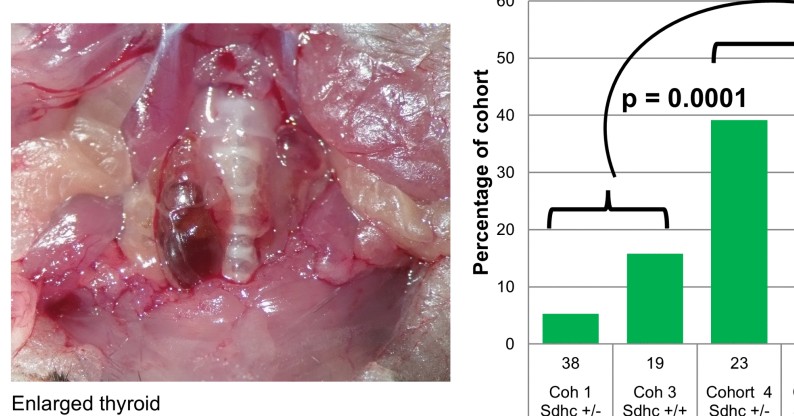

Enlarged thyroid

**Fig 6. Thyroid abnormalities are common in Sdhc+/- Rb1:7 cohorts.** Mouse 9104481 (2.5X objective) showing an enlarged thyroid, visible next to the trachea. In general, gross pathology of the thyroid was rare in the C57BL6 mouse background, as illustrated by the Sdhc+/- cohort (5%, left panel). Bars show percentage of each cohort. Numbers of mice shown below bars. Cross-cohort Chi Sq significance p = 0.006 (corrected). Cohorts 1-3 versus 4-7, p = 0.0001 (corrected). Cohorts 4&5 versus 6&7, p = NS. Cohorts 4&6 (maternally transmitted mutation) versus 5&7 (paternally transmitted mutation), p = NS. Corrected = correction for multiple testing (13 tests).

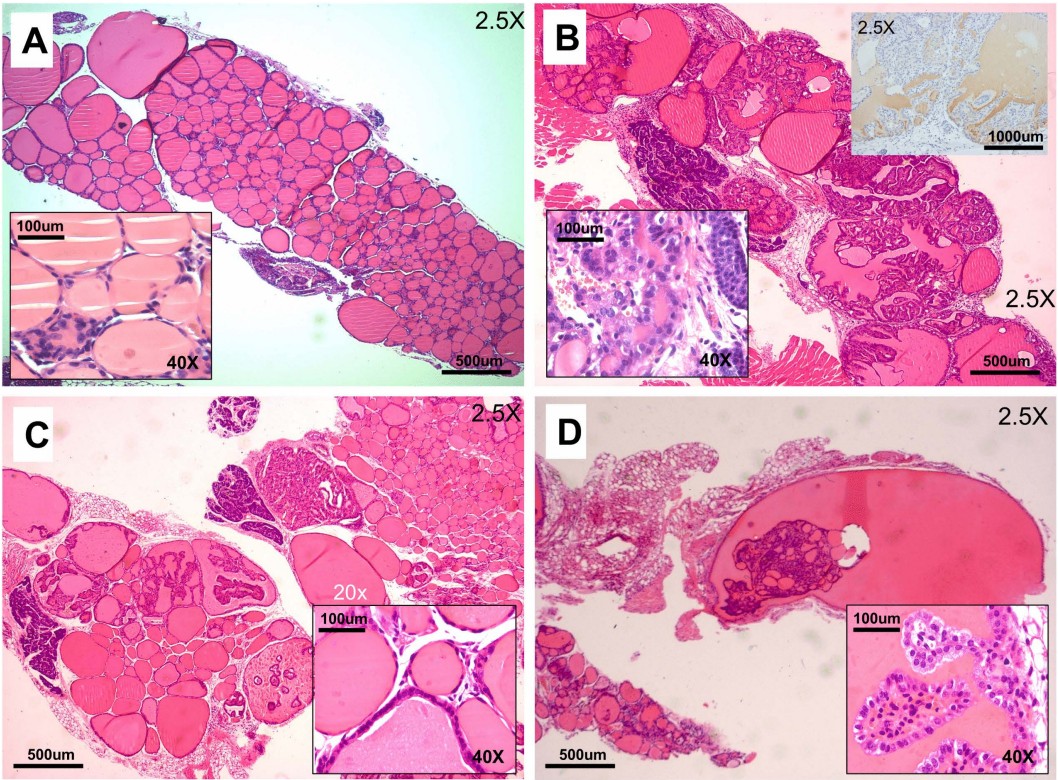

**Fig 7. Histology of typical thyroid abnormalities.** (A) Hematoxylin-eosin histology of thyroid from a healthy wildtype female mouse; 2.5x objective. Inset 40x objective. Diagnosis based on morphology and/or staining for thyroglobulin. (B) Thyroid from mouse 9098841 (female, cohort 5 or 7 – metaphase unsuccessful) showing highly abnormal histology, diagnosed as a papillary carcinoma or hyperplasia; 2.5x objective. Inset lower left H&E 40x objective; inset upper right 2.5x objective, anti-thyroglobulin IHC. (C) Thyroid from 9103313 (female, cohort 7) diagnosed as a papillary carcinoma; 2.5x objective. Inset 40x objective. (d) 9104481 (female, cohort 7) diagnosed as a papillary carcinoma; 2.5x objective. Inset 40x objective.

## Discussion

This study had two primary goals: 1) to test the Hensen model of paternal inheritance of SDH-related tumour risk, and 2) to develop a paraganglioma mouse model to study the genesis of paraganglioma. While not confirmed in this study, neither is the Hensen model refuted, and it remains the most compelling explanation of the available evidence on Sdh paternal inheritance [18]. Numerous groups over the past two decades have sought to create an Sdh-related paraganglioma mouse model, but none have been successful to date [19]. A third goal of this study was, 3) to explore the feasibility of investigating genetic modifiers using non-standard chromosomal configurations in the mouse. In this case, the objective was partly successful, in that a combination of a heterozygous genetic knockout and a Rb chromosome produced a stronger phenotype compared to the Rb/Rb or *Sdhc+/-* cohorts alone (50% vs. 15% or 5%).

### A human semi-homologous chromosome in mice

Rb translocations in mice have been studied extensively but mainly in relation to processes that result in chromosomal races, reproductive isolation and speciation. Phenotypically, Rb chromosomes are sometimes associated with reduced fertility [15], and have been associated with differences in growth of the mouse mandible [20], but few other phenotypic traits of wild or laboratory-housed Rb mice have been reported. Karyotype analyses of human *SDHD* and *SDHAF2*

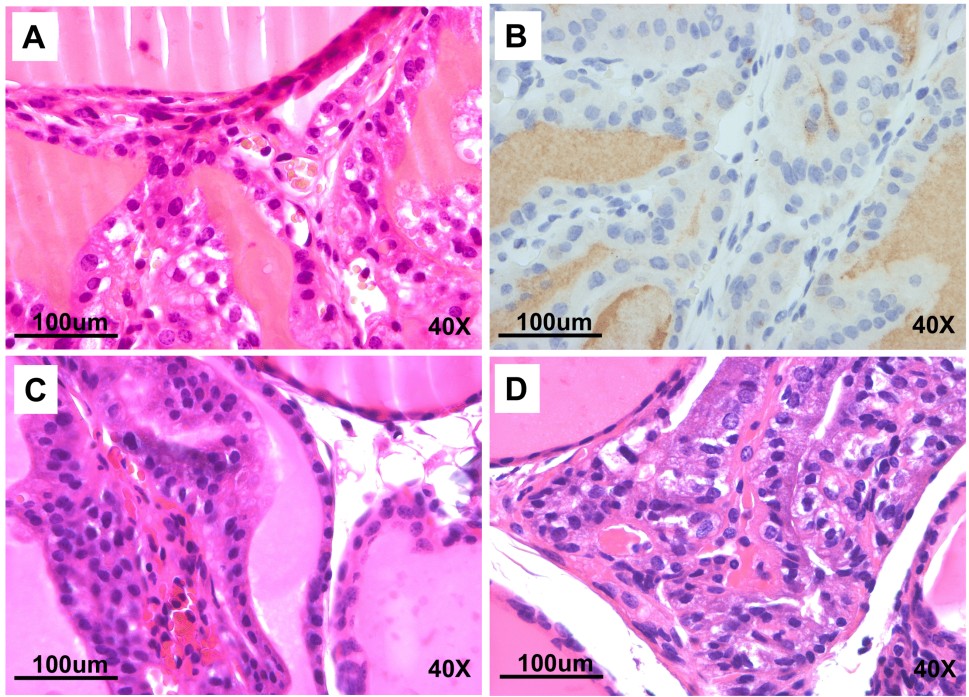

**Fig 8. Detailed histology of thyroid abnormalities found in cohort 4 to 7 mice. (A)** Thyroid from mouse 9098841 (female, cohort 5 or 7 – metaphase unsuccessful) showing highly abnormal histology, including enlarged nuclei and diagnosed as a papillary carcinoma or hyperplasia. **(B)** Thyroid from mouse 9098841 stained using an anti-thyroglobulin antibody. **(C)** Thyroid from 9101254 (female, cohort 5) diagnosed as possible C cell proliferation. **(D)** Thyroid from 9103110 (female, cohort 5) diagnosed as cystic papillary carcinoma.

tumours characteristically show loss of the maternal chromosome [8–10,21] and the few cases of maternal inheritance described show mitotic recombination and subsequent loss of the short arm of maternal chromosome 11 [22,23], underlining the essential role of this chromosomal region in tumourigenesis. The strategy adopted in the present study was chosen because the modifier required for tumour initiation in human carriers of *SDHD* and *SDHAF2* has not been precisely defined [24]. It is also possible that more than one gene is involved, at least one of which is likely imprinted on the paternal chromosome and expressed from the maternal chromosome. While our strategy brought together homologous chromosomal regions essential for SDH-initiated tumourigenesis, a weakness of this approach is that certain homologous regions are absent and in the event of loss of the maternal chromosome, non-homologous regions will be lost that could potentially cause or inhibit emergence of phenotypes.

## Rodent models of PPGL

The effort to develop rodent models of SDH-linked disease has a long history, stretching back to 2004 when Piruat et al. described a systemic knockout of *Sdhd* in the mouse [25]. Our group described a similar *Sdhd* knockout, as well as in combination with H19 as a putative modifier gene of paternal inheritance [13]. Even followed for their full lifespan, none of these mice developed PPGL. Subsequent studies encompassing *Sdhb*, *Sdhc* and *Sdhd* utilized conditional models with tissue-specific promoters. None resulted in PPGL (reviewed by Lussey-Lepoutre et al. [19]. Why didn't this model result in PPGL, and is there a future for SDH-related tumour models? In addition to the weaknesses discussed above, the model described here utilized *Sdhc* rather than *Sdhd* or *Sdhaf2*, which may have influenced outcomes. As pheochromocytomas (adrenal paragangliomas) are relatively common in various mouse and rat models, straightforward explanations such

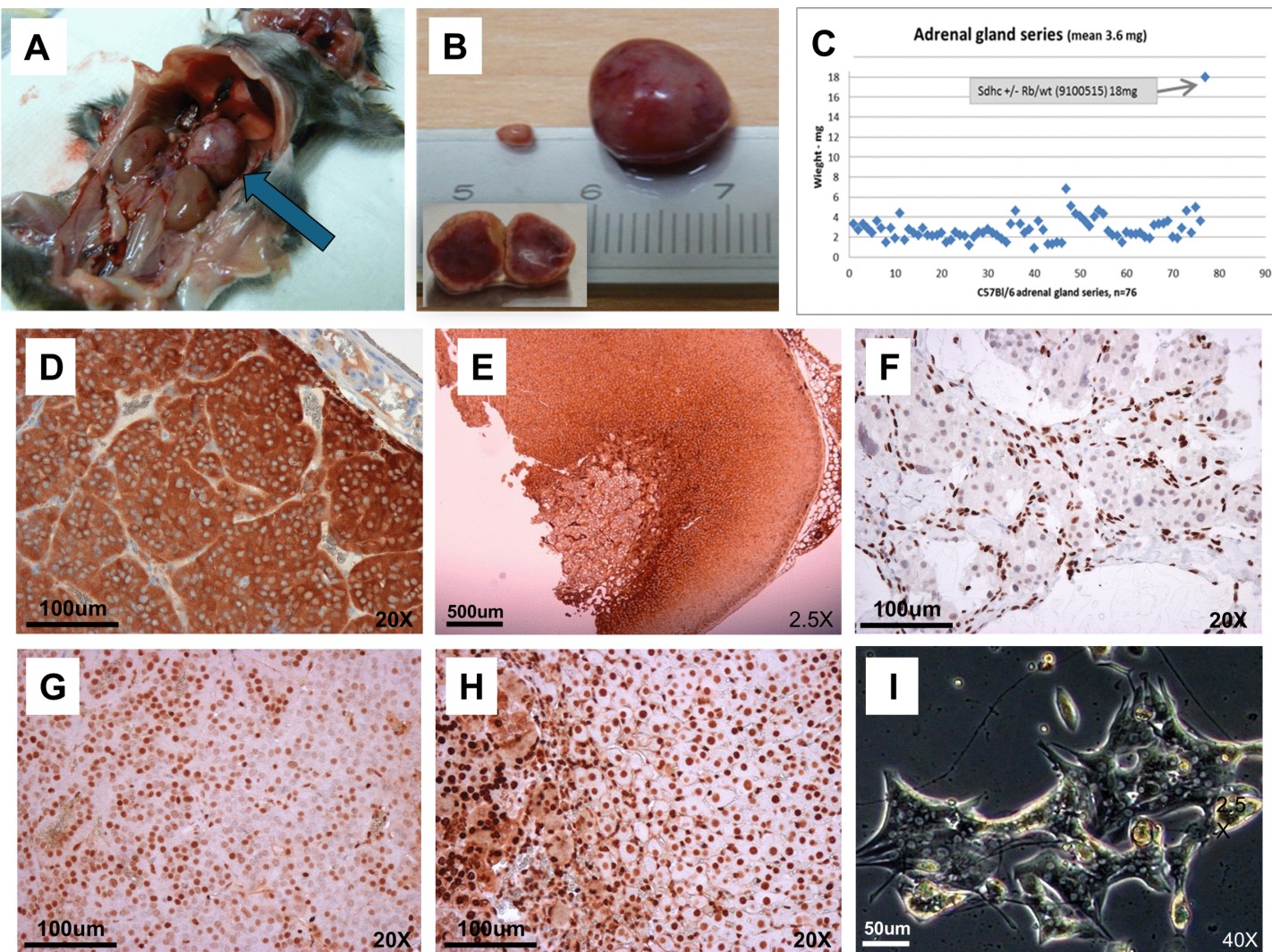

**Fig 9. Bilateral pheochromocytoma.** (A) Mouse 9100515 Sdhc +/- Rb/? (female, cohort 5 or 7 – metaphase unsuccessful) shown with in situ left-sided pheochromocytoma. The enlarged right-sided adrenal is also visible. (B) The right-sided adrenal weighted 18 mg and the left-sided 643 mg. Inset shows cross-section of the larger adrenal. (C) A series of 76 adrenal glands from mice with a C57BL/6 background. At 18 mg the right-sided adrenal was a major outlier. (D) Anti-tyrosine hydroxylase immunohistochemical staining, a marker protein found in the adrenal medulla and pheochromocytomas, shows typical histology in the 9100515 pheochromocytoma. (E) Anti-SDHB staining is generally negative in SDH-associated tumours due to loss of the SDH complex, but is positive here, suggesting a tumour unrelated to Sdhc loss. (F) Another characteristic of SDH-deficient tumors is loss of 5-hydroxymethylcytosine (5hmC), as seen here in a human SDHB-deficient tumour. (G) Anti-5hmC staining of 9100515 pheochromocytoma cells compared with a wildtype mouse (h) showed retention of 5hmC. (i) 9100515 pheochromocytoma cells in culture showed unusual morphology but failed to proliferate.

as lifespan or tissue-specific capacity for tumorigenesis seem unlikely. In humans, loss of a still unidentified modifier(s) appears essential for *SDHD* and *SDHAF2* tumour initiation [10]. Forced deletion of this modifier in the mouse may be the key to initiating PPGL in this species. However, we and others have shown that loss of chromosome 11 occurs but is not a signature feature of human *SDHB*-related PPGL [10,12], so this explanation does not explain the failure of *Sdhb* mouse models, unless that gene requires loss of an independent modifier in order to initiate tumourigenesis. A more likely explanation for the failure of this and all previous models is biochemical variance between mouse and man. Paragangliomas

are commonly reported in only two species, humans and canines. During co-evolution with humans, dogs developed an altered carbohydrate metabolism [26,27], so some specific feature of carbohydrate metabolism that interacts with succinate dehydrogenase may create a biochemical environment conducive to PPGL formation. This idea has not to our knowledge been investigated to date.

### Inflammation

A novel observation in this study was the high incidence of gross inflammation in mice carrying only one copy of an Rb(1:7) chromosome. Cohorts 4 and 5, which carried one Robertsonian (metacentric) 1:7 chromosome (together with telocentric chromosomes 1 and 7), showed a significantly higher frequency of gross inflammation upon necropsy compared to *Sdhc+/-* mice carrying two Rb(1:7) chromosomes or the control cohorts. The absence of similar findings in the Rb/Rb cohorts 6 and 7 suggests that *Sdhc* did not play a role in this phenotype. Heterozygosity for a metacentric chromosome has previously been reported to influence some phenotypes [28], but immune anomalies have not been reported to our knowledge.

### Thyroid

With an incidence of 1%, thyroid carcinoma is the most common endocrine cancer and the subtype papillary thyroid carcinoma (PTC) is the most frequent variety, accounting for around 80% of all thyroid cancers. PTCs are a minor diagnostic criterium of Cowden Syndrome and have been reported together with PPGL [29,30], and together with *SDHB* [31], *SDHC* [32–34], *SDHD* [35–38] and *SDHAF2* [39] gene variants. The lower level of thyroid abnormalities in *Sdhc+/-* mice (Cohort 1, Figs 2 and 6) compared to Rb/Rb *Sdhc+/+* (Cohort 3, Figs 2 and 6) suggest that the Robertsonian chromosome was the primary contributor to the phenotype, but the high levels in the combined Rb/wt *Sdhc+/-* and Rb/Rb *Sdhc+/-* cohorts (Cohorts 4–7, Figs 2 and 6) indicate potent synergy. We did not further explore the molecular basis of this finding, so cannot comment on whether it was associated with chromosomal loss or was a consequence of altered gene expression from metacentric Robertsonian versus telocentric chromosomes.

To explore the association of succinate dehydrogenase (SDH) to human thyroid cancer, Ashtekar and colleagues generated mice lacking *Sdhd* in the thyroid. These mice developed enlarged thyroid glands with follicle hypercellularity and increased proliferation. In addition, knockdown of *SDHD* in human thyroid cell lines caused enhanced migratory capability and stem-like features, which were also observed in the mouse tumours. These characteristics could be reversed by α-ketoglutarate, suggesting dedifferentiation occurs due to an imbalance in TCA metabolites [40]. In a study focussed on Cowden Syndrome patients, Ni and colleagues found that both papillary and follicular thyroid tumours showed consistent loss of *SDHC/SDHD* gene expression, which was associated with earlier disease onset and higher pathological-TNM stage [34].

### Pheochromocytoma

Another unusual finding of the study was the occurrence of a pheochromocytoma in a mouse from the experimental cohort 5/7 (metaphase unsuccessful, thus the mouse was Rb/wt or Rb/Rb), the experimental cohorts with paternally inherited *Sdhc* loss designed to directly mimic paternal inheritance in the Hensen model. While many rodent models develop adrenal pheochromocytomas, including Nf1 knockouts [41], c-Mos transgenics [42], RET Met918 transgenics [43], Cdkn1b-mutated Sprague–Dawley rats [44], Rb1/Trp53 dual knockouts [45], ceramide synthase 2 knockout mice [46], ErbB2 transgenics [47], connexin 32 knockouts [48], PTEN knockouts [49] and B-Raf transgenics [50], spontaneous pheochromocytoma in the C57BL/6 mouse background has not been reported to our knowledge. The co-occurrence of a pheochromocytoma with a smaller tumour/hyperplasia therefore suggests a germline or very early genetic event. However, loss of *Sdhc* did not appear to play a role.

## Conclusion

This study did not re-capitulate features of the Hensen Model, and the phenotypes noted are not characteristic of this model. As both the present mouse model as well as our previous *Sdhd*/H19 KO did not develop paragangliomas, the Hensen model remains hypothetical. A conclusive test of the Hensen model will likely require an animal model that regularly develops paraganglioma. The striking phenotypes reported here also highlight the lack of Robertsonian studies to date that considered phenotype or pathology. We conclude that chromosomal constitution can dramatically impact phenotype. The approach chosen here, if more widely applied, may aid in the dissection of other diseases in which modifiers are suspected.

## Acknowledgments

We thank Professor Judith Bovee for assistance with thyroglobulin staining, and Professor Hans Morreau and Dr. Daniela Salvatori for assessment of thyroid pathology.

## Author contributions

**Conceptualization:** Jean-Pierre Bayley, Peter Devilee.

**Data curation:** Jean-Pierre Bayley.

**Formal analysis:** Jean-Pierre Bayley, Heggert G. Rebel.

**Funding acquisition:** Jean-Pierre Bayley.

**Investigation:** Jean-Pierre Bayley, Heggert G. Rebel.

**Methodology:** Jean-Pierre Bayley, Peter Devilee.

**Project administration:** Jean-Pierre Bayley.

**Resources:** Jean-Pierre Bayley.

**Supervision:** Jean-Pierre Bayley, Peter Devilee.

**Validation:** Jean-Pierre Bayley.

**Visualization:** Jean-Pierre Bayley.

**Writing – original draft:** Jean-Pierre Bayley.

**Writing – review & editing:** Jean-Pierre Bayley, Peter Devilee.

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
