## [Decision Letter · Decision Letter 0]

2 Feb 2026

Dear Dr. Bayley,

Thank you for submitting your manuscript to PLOS ONE. After careful consideration, we feel that it has merit but does not fully meet PLOS ONE’s publication criteria as it currently stands. Therefore, we invite you to submit a revised version of the manuscript that addresses the points raised during the review process.

We look forward to receiving your revised manuscript.

Kind regards,

Avaniyapuram Kannan Murugan, M.Phil., Ph.D.

Academic Editor

PLOS One

**Journal Requirements:**

1. When submitting your revision, we need you to address these additional requirements. Please ensure that your manuscript meets PLOS ONE's style requirements, including those for file naming. The PLOS ONE style templates can be found at https://journals.plos.org/plosone/s/file?id=wjVg/PLOSOne_formatting_sample_main_body.pdf and https://journals.plos.org/plosone/s/file?id=ba62/PLOSOne_formatting_sample_title_authors_affiliations.pdf 2. To comply with PLOS ONE submissions requirements, in your Methods section, please provide additional information regarding the experiments involving animals and ensure you have included details on (a) methods of sacrifice, and (b) efforts to alleviate suffering. 3. We note that the grant information you provided in the ‘Funding Information’ and ‘Financial Disclosure’ sections do not match.  When you resubmit, please ensure that you provide the correct grant numbers for the awards you received for your study in the ‘Funding Information’ section. 4. Thank you for stating the following financial disclosure: Dutch Cancer Foundation (KWF UL 2011-5025) and The Paradifference Foundation   Please state what role the funders took in the study.  If the funders had no role, please state: "The funders had no role in study design, data collection and analysis, decision to publish, or preparation of the manuscript." If this statement is not correct you must amend it as needed. Please include this amended Role of Funder statement in your cover letter; we will change the online submission form on your behalf. 5. We note that your Data Availability Statement is currently as follows: All relevant data are within the manuscript and its Supporting Information files. Please confirm at this time whether or not your submission contains all raw data required to replicate the results of your study. Authors must share the “minimal data set” for their submission. PLOS defines the minimal data set to consist of the data required to replicate all study findings reported in the article, as well as related metadata and methods (https://journals.plos.org/plosone/s/data-availability#loc-minimal-data-set-definition). For example, authors should submit the following data: - The values behind the means, standard deviations and other measures reported;- The values used to build graphs;- The points extracted from images for analysis. Authors do not need to submit their entire data set if only a portion of the data was used in the reported study. If your submission does not contain these data, please either upload them as Supporting Information files or deposit them to a stable, public repository and provide us with the relevant URLs, DOIs, or accession numbers. For a list of recommended repositories, please see https://journals.plos.org/plosone/s/recommended-repositories. If there are ethical or legal restrictions on sharing a de-identified data set, please explain them in detail (e.g., data contain potentially sensitive information, data are owned by a third-party organization, etc.) and who has imposed them (e.g., an ethics committee). Please also provide contact information for a data access committee, ethics committee, or other institutional body to which data requests may be sent. If data are owned by a third party, please indicate how others may request data access. 6. We note that you have included the phrase “data not shown” in your manuscript. Unfortunately, this does not meet our data sharing requirements. PLOS does not permit references to inaccessible data. We require that authors provide all relevant data within the paper, Supporting Information files, or in an acceptable, public repository. Please add a citation to support this phrase or upload the data that corresponds with these findings to a stable repository (such as Figshare or Dryad) and provide and URLs, DOIs, or accession numbers that may be used to access these data. Or, if the data are not a core part of the research being presented in your study, we ask that you remove the phrase that refers to these data. 7. Please amend either the abstract on the online submission form (via Edit Submission) or the abstract in the manuscript so that they are identical. 8. Your ethics statement should only appear in the Methods section of your manuscript. If your ethics statement is written in any section besides the Methods, please move it to the Methods section and delete it from any other section. Please ensure that your ethics statement is included in your manuscript, as the ethics statement entered into the online submission form will not be published alongside your manuscript. 9. Please upload a new copy of Figures 1 and 2, as the detail is not clear. Please follow the link for more information:  https://journals.plos.org/plosone/s/figures 10. If the reviewer comments include a recommendation to cite specific previously published works, please review and evaluate these publications to determine whether they are relevant and should be cited. There is no requirement to cite these works unless the editor has indicated otherwise. 

**Additional Editor Comments:**Address all the comments carefully, particularly Reviewer 1. Any unaddressed comments will result in re-revision.

Reviewers' comments:

Reviewer's Responses to Questions

**Comments to the Author**

1. Is the manuscript technically sound, and do the data support the conclusions?

Reviewer #1: Yes

Reviewer #2: Yes

2. Has the statistical analysis been performed appropriately and rigorously?

Reviewer #1: Yes

Reviewer #2: Yes

3. Have the authors made all data underlying the findings in their manuscript fully available?

Reviewer #1: Yes

Reviewer #2: Yes

4. Is the manuscript presented in an intelligible fashion and written in standard English?

Reviewer #1: Yes

Reviewer #2: Yes

**Reviewer #1:**  This study presents a genetically elegant and technically demanding strategy combining an Sdhc knockout allele with a Robertsonian (Rb(1:7)) fusion chromosome to interrogate potential modifier effects and parent-of-origin mechanisms relevant to SDH-associated tumorigenesis. Although the model does not recapitulate PPGL, the emergence of thyroid lesions with PTC-like morphology is unexpected and scientifically interesting. Overall, the study is scientifically solid and methodologically sound, and it provides valuable data despite not fully achieving the intended tumor phenotype. I believe the manuscript is suitable for publication after the authors address the points below. This study presents a genetically elegant and technically demanding strategy combining an Sdhc knockout allele with a Robertsonian (Rb(1:7)) fusion chromosome to interrogate potential modifier effects and parent-of-origin mechanisms relevant to SDH-associated tumorigenesis. Although the model does not recapitulate PPGL, the emergence of thyroid lesions with PTC-like morphology is unexpected and scientifically interesting. Overall, the study is scientifically solid and methodologically sound, and it provides valuable data despite not fully achieving the intended tumor phenotype. I believe the manuscript is suitable for publication after the authors address the points below.

Specific Comments

• The central mechanistic assumption—that whole-chromosome loss contributes to the observed synergy—remains inferential. Even limited validation (FISH, CNV, allele-specific PCR) in a subset of samples would strengthen the conclusions. If additional experiments are not feasible, please explicitly acknowledge this limitation and discuss alternative explanations.

• Since the study references the Hensen hypothesis and parent-of-origin effects, the manuscript should clearly state whether the Sdhc knockout allele was inherited paternally or maternally, whether reciprocal crosses were performed or considered, and how this influences interpretation.

• The thyroid lesions resemble papillary thyroid carcinoma, but molecular characterization is limited. Please temper the terminology by using phrases such as “PTC-like features” or “PTC-like morphology” unless molecular markers (e.g., TTF1, PAX8, BRAF) can be provided.

• A brief and balanced discussion of why this model did not generate PPGL, and how this informs future SDH-related tumor models, would improve the clarity of the Discussion section.

• Minor issues include improving the resolution or annotation of some histology figures, ensuring consistent gene nomenclature for mouse vs. human, clarifying sample sizes in figure legends, and minor proofreading. A short comparison with previously published SDH mouse models would further help contextualize the novelty of the work.

**Reviewer #2:** [1] How to perform immunohistochemistry? Description of immunohistochemistry is required (Figure 9g).[1] How to perform immunohistochemistry? Description of immunohistochemistry is required (Figure 9g).

[2] Why PCR data is not available?

[3] It is required to show scale bars for Figure 7 to Figure 9; magnification is not enough.

[4] Current challenge would be, an establishment of mouse model to study paraganglioma, and authors partially achieved that goal. However, there is no clear statement of perspective directions. Having this work, how authors would investigate?

**Do you want your identity to be public for this peer review?** For information about this choice, including consent withdrawal, please see our For information about this choice, including consent withdrawal, please see our Privacy Policy .

Reviewer #1: No

Reviewer #2: No

---

## [Author Response · Author response to Decision Letter 1]

3 Mar 2026

Reviewer #1: This study presents a genetically elegant and technically demanding strategy combining an Sdhc knockout allele with a Robertsonian (Rb(1:7)) fusion chromosome to interrogate potential modifier effects and parent-of-origin mechanisms relevant to SDH-associated tumorigenesis. Although the model does not recapitulate PPGL, the emergence of thyroid lesions with PTC-like morphology is unexpected and scientifically interesting. Overall, the study is scientifically solid and methodologically sound, and it provides valuable data despite not fully achieving the intended tumor phenotype. I believe the manuscript is suitable for publication after the authors address the points below.

Specific Comments

• The central mechanistic assumption—that whole-chromosome loss contributes to the observed synergy—remains inferential. Even limited validation (FISH, CNV, allele-specific PCR) in a subset of samples would strengthen the conclusions. If additional experiments are not feasible, please explicitly acknowledge this limitation and discuss alternative explanations.

Reply: We refer to a mechanistic assumption underlying the Hensen Model, which is based on five publications that all showed loss of chromosome 11 in paragangliomas. These references [5-10] can be found on line 71. However, these findings do not appear relevant to the current data as we did not observe the expected phenotype. We used the word ‘synergy’ once in the manuscript, referring to thyroid abnormalities. This was based on differences in the frequency of thyroid abnormalities across cohorts and implied an interaction between the Rb chromosome and heterozygous loss of Sdhc via KO. This is not the same as chromosome loss, for which we have no evidence here. We have added the following, line 443: “We did not further explore the molecular basis of this finding, so cannot comment on whether it was associated with chromosomal loss or was a consequence of altered gene expression from metacentric Robertsonian versus telocentric chromosomes”.

• Since the study references the Hensen hypothesis and parent-of-origin effects, the manuscript should clearly state whether the Sdhc knockout allele was inherited paternally or maternally, whether reciprocal crosses were performed or considered, and how this influences interpretation.

Reply: We previously stated “These crosses ensured inheritance of Sdhc via either the female (4&6) or male (5&7) line in order to model human patterns of inheritance. The two cohorts carrying the paternally-transmitted Sdhc KO (4&6) simulate patients at risk of tumour development due to a paternally-inherited SDHD or SDHAF2 variant (Figure 1c)”. We have now made this more explicit from line 203 onwards.

• The thyroid lesions resemble papillary thyroid carcinoma, but molecular characterization is limited. Please temper the terminology by using phrases such as “PTC-like features” or “PTC-like morphology” unless molecular markers (e.g., TTF1, PAX8, BRAF) can be provided.

Reply: Indeed. We have now modified the title and text to better acknowledge this shortcoming.

• A brief and balanced discussion of why this model did not generate PPGL, and how this informs future SDH-related tumor models, would improve the clarity of the Discussion section.

Reply: The million-dollar question! We have now added a section to the discussion (Rodent models of PPGL) from line 398. Our guess: it’s the biochemistry.

• Minor issues include improving the resolution or annotation of some histology figures, ensuring consistent gene nomenclature for mouse vs. human, clarifying sample sizes in figure legends, and minor proofreading. A short comparison with previously published SDH mouse models would further help contextualize the novelty of the work.

Response:

(i) “resolution or annotation of some histology figures”; Reply: Looking through the figures they appear to us of adequate or even good resolution. We have reworked most of the figures.

(ii) “consistent gene nomenclature for mouse vs. human” ; Reply: we have now checked and corrected these inconsistencies.

(iii) “clarifying sample sizes in figure legends” ; Reply: we have now added actual mouse numbers to figures 3, 5 and 6.

(iv) “minor proofreading” ; Reply: we have carefully edited the manuscript.

(v) “short comparison with previously published SDH mouse models” ; Reply: We have now added a section to the discussion (Rodent models of PPGL) from line 398.

Reviewer #2: [1] How to perform immunohistochemistry? Description of immunohistochemistry is required (Figure 9g).

Reply: Apologies for this oversight. A section has now been added to Methods.

[2] Why PCR data is not available?

Reply: The gels were of poor quality and not fit for display. We have therefore removed the sentence referring to this data.

[3] It is required to show scale bars for Figure 7 to Figure 9; magnification is not enough.

Reply: This has now been corrected.

[4] Current challenge would be, an establishment of mouse model to study paraganglioma, and authors partially achieved that goal. However, there is no clear statement of perspective directions. Having this work, how authors would investigate?

Reply: The million-dollar question! We have now added a section to the discussion (Rodent models of PPGL) from line 398. Our guess: it’s the biochemistry.

---

## [Editor Report · Decision Letter 1]

10 Mar 2026

A novel genetic strategy to interrogate an unknown phenotypic modifier: Sdhc KO-Robertsonian mice develop frequent thyroid abnormalities with papillary thyroid carcinoma-like features

PONE-D-25-58546R1

Dear Dr. Bayley,

We’re pleased to inform you that your manuscript has been judged scientifically suitable for publication and will be formally accepted for publication once it meets all outstanding technical requirements.

Kind regards,

Avaniyapuram Kannan Murugan, M.Phil., Ph.D.

Academic Editor

PLOS One
---

## [Editor Report · Acceptance letter]

PONE-D-25-58546R1

PLOS One

Dear Dr. Bayley,

I'm pleased to inform you that your manuscript has been deemed suitable for publication in PLOS One. Congratulations! Your manuscript is now being handed over to our production team.

Kind regards,

on behalf of

Dr. Avaniyapuram Kannan Murugan

Academic Editor

PLOS One